# OpenReview forum: "GLOSE: Understanding and Mitigating Toxicity in LLMs via Global Toxic Subspace"
_ICLR.cc/2026/Conference — ICLR 2026 Conference Withdrawn Submission_

### Official Review · Reviewer_bqZq · 2025-10-21

**Soundness:** 1
**Presentation:** 1
**Contribution:** 2
**Rating:** 2
**Confidence:** 4

**Summary:**

The paper proposes an algorithm, GLOSE for mitigating toxic language generation in LLMs. A previous work, ProFS (Uppaal et al., 2025), had already shown that toxicity can be mitigated by removing layer-wise subspaces responsible for toxic language. GLOSE builds on ProFS by estimating a global shared subspace for the whole model. The authors show on two toxic inducing datasets that the proposed method achieves better mitigation than ProFS and other baselines on a variety of models while minimally degrading fluency, perplexity and consistency.

GLOSE first computes the candidate directions for each layer by applying SVD to the activation differences between toxic and non-toxic input pairs. Then all directions are decoded into a vocabulary and are ranked based on how often they produce a token that belongs to a list of “bad words”. All directions (across the model) above a given threshold are used to form a new matrix from which principal component are extracted and used to “identify” the global toxic subspace of the model. Finally the global subspace is removed by projecting the existing weights onto the orthogonal complement of this toxic global subspace.

**Strengths:**

According to the results presented, the proposed algorithm achieve good mitigation at low cost of fluency, perplexity and consistency. The paper also tackles an important problems for the community.

**Weaknesses:**

While the results are favorable to the proposed methods I have a variety of concerns. Roughly in order of importance:  the experimental protocol is not complete, some important details are not stated and some metrics and baselines are missing, there are unsupported claims and unclear conclusions from the results presented, and lastly the writing, it is not always clear and there are a few errors the hinder the understanding of the method. Allow me to elaborate on all the above.

Experimental results:
- From the paper it is not clear which dataset is used to construct the subspace. If the subspace was estimated using one dataset, and  results on mitigation were on a different dataset that would align with how mitigation was assessed in previous work (e.g., LineAS by Rodriguez at al. 2025). However, since it is not clear which dataset was used for the subspace estimation I fear that the authors used the same dataset used for testing. If this was the case then one would have to show that the method generalize to other datasets.
- Missing MMLU. MMLU is a must for these type of work to show that the model abilities have not been degraded.
- Missing baselines. Some of the neuron based steering baselines would have strengthen the paper. Especially considering that some of those work with only 64 sentences (32 toxic and 32 non-toxic) while the current method used 500.
- While the results are mostly favorable there isn’t any single standard deviation reported. Hence it is impossible to assess their significance.
- Minor:
    - I appreciate the effort to present ablation studies. However when evaluating the ranking step you have shown that ranking is better than random projections but it would be stronger to show that your form of ranking is better than a simpler one (e.g., using the eigenvalues as “score”).
    - All hyper parameters are listed in Table 5 but I believe the number  “m” for the top-m tokens decoded during the ranking stage is missing.
    - In ProFS they use a mean projection to avoid editing stop-words. You just mention that you perform mean-centering but it is not clear if this is the same procedure or if you compute the mean across all positive and negative sets?
    - How are $X^+^ and $X^-$ computed over the sequence length? Do you average or use last token? This should be explained.

Unsupported Claims
- The main unsupported claim is that the method “achieves state-of-the-art detoxification while preserving general capabilities.” The most important test used in SOTA to show general capabilities (MMLU) is not presented. Fluency, Perplexity and consistency are great but far less convincing that MMLU. This evaluation without MMLU is incomplete.
- Statements like “leaving them vulnerable to adversarial attacks that reactivate toxicity” are not supported. I did not see experiments that showed that “toxicity” was “reactivated”. I am also not 100% sure about what it means.
- Another unsupported claim is “we discover that each LLM contains a global toxic subspace across layers.” The authors mechanically build one why is this a “discovery” ?
- Through out the paper the subspace is called “global” but from Table 5 in the Appendix it seems that the “global” subspace is only extracted from a some specific layers rather than the whole model.
- Of the two identified:
    - (1) is it a practical limitation? Can the end user leverage this weakness for an attack?
    - (2) What does this mean?? Not clear from the abstract. Contrastive objective over limited samples inject noise into layer-wise subspaces, hindering stable extraction.

Unclear conclusions (or not well explained conclusions) drawn from the experiments. E.g.,
- In section 3.1 when discussing the ”LIMITATIONS OF TOXIC VECTORS” 3 results are presented that show that both sign and magnitude of the activations play a role in producing toxic output. I would say that these results are expected, but more importantly it is not clear why from these results the authors conclude that “binary classification is misleading since toxicity also depends on activation state” and hence this is a limitation of methods based on vectors. As far as I understand methods based on vectors identify a boundary that divides the space into two (hence binary) hyperplanes (e.g., ITI). Assuming this is a good boundary (i.e., well identified), the further one goes from this boundary (magnitude) the more or less toxic (depending on the sign) the generated language. This seems a known fact, unless the authors meant something different, which is not obvious from the way it is explained.
- In section 3.2 when discussing the “LIMITATIONS OF LAYER-WISE TOXIC SUBSPACE” the authors identify the top directions at each layers and decode them into tokens finding that “projections from middle layers predominantly yield toxic tokens, while those from lower and upper layers do not exhibit this pattern” and from this the authors conclude that “This confirms that layers have different toxic modeling capacities, making layer-wise toxic subspaces inconsistent and unreliable.” I would have concluded the opposite: since different layers produce different levels of toxicity having a per-layer intervention would make sense.
- The authors then use the two conclusions from above to justify the exploration of the use of a global subspace. However, since none of the two conclusions above is convincing (or perhaps not clearly explained) this leap seems unjustified.
- Results in Figure 3 show that the toxic directions have high cosine similarity mostly between neighbor layers and they don’t seem at all similar across the whole model. From these findings again the authors deduce that “toxic regions in FFN layers are best characterized by a global toxic subspace formed through the orthogonal combination of multiple shared toxic directions, rather than isolated layer-specific vectors or single directional biases”. However doesn’t this result simply indicate again that per-layer subspace makes more sense?
- Similarly, another conclusion that I found unclear was that because “FFN exhibit varying capacity for toxicity”  therefore  “It makes layer-wise extraction methods susceptible to noise interference from limited samples, hindering stable extraction of layer-wise toxic subspaces”.

Writing
- The paper is missing a “Related work section”. While there is a small discussion in the introduction this is not sufficient. For example, the whole field and recent work on neuron activation steering is completely missing from the review and the baselines. A few work within this category (but there are be more, these are just examples) that I was expecting to find are  ITI (Li et al., 2024),  Lin-AcT (Rodriguez et al., 2025), RepE, by Zou et al. (2023).
- The symbol $\alpha$ is used three times, in different contexts. Once in Equation 3 to quantify the layer’s capacity for toxic expression,  in equation 4 as an heuristic scaling factor, and finally in equation 7 to control the “selection strictness”
- Minor:
    - English could be improved, a few typos should be corrected (even the method is sometimes called GLOSE and others GloSS).
    - The metrics used in the main sections should be explained in the main section not in the appendix.
    - Where does eq 3 comes from? Also that factorization seems to have been introduced by Uppaal et al., 2025  so this should be made more explicit.
    - From the abstract the authors identify 2 “critical limitations: (1) Removed toxic vectors can be reconstructed via linear combinations of non-toxic vectors”, and (2) Contrastive objective over limited samples inject noise into layer-wise subspaces, hindering stable extraction.” It is not clear if (1) is just a theoretical limitation or fi this reconstruction happens in practice, and (2) it is not clear why a per-layer subspace estimation should be hindered more than a global estimation given the same number of samples. Shouldn’t that be the way around?

Finally a general weakness (however this is often in common with other methods so I am mentioning it here just to raise awareness) is that this method is based  on “bad language” rather than toxic language. In this case perhaps more than other as there is an explicit step for the ranking of the directions where only directions that are decoded into “bad words” are considered. However, toxic language goes well beyond bad words, and arguably the most dangerous toxic language is that that does not use vulgar language (which is arguably easy to detect).

**Questions:**

I am afraid that there are too many concerns to address in a rebuttal. However I remain open to the fact that I might have fundamentally misunderstood this work and none of my concerns above are motivated.

---

### Official Review · Reviewer_D7k2 · 2025-10-24

**Soundness:** 2
**Presentation:** 2
**Contribution:** 2
**Rating:** 2
**Confidence:** 5

**Summary:**

This work proposes to estimate a global (for all layers) toxic subspace in activation space, and then project the model weights onto it to effectively remove toxicity. Experiments on 6 modern LLMs are provided, analyzing toxicity (2 datasets) and utility (3 metrics). The experiments show that the proposed approach is effective, and preserves utility to some extent. Additional experiments on the transferability of steering vectors across layers and the effect of positive and negative steering are also provided, as well as an ablation study for the method steps.

**Strengths:**

**S1:** This work addresses the important topic of toxicity mitigations, and raises the question whether there exists a global toxic subspace for all layers.

**S2:** The suite of models used is of interest, and the metrics used are sensible.

**Weaknesses:**

**W1:** _We test comprehensive control by defining $W_{toxic}$ as the toxic direction and implementing two steering strategies: toward toxic direction (preserving activation signs based on cosine similarity) and away from toxic direction (flipping all activation signs). _

Using $W_{toxic}$ to determine the steering direction has already been proposed in the past. I encourage the authors to cite ITI [Kenneth et al. NeurIPS 2024](https://arxiv.org/abs/2306.03341). Moreover, studies showing the impact of steering towards/against also exist, for example in CAA (Panickssery et al. 2024).

_These results demonstrate that activation sign and magnitude critically determine toxic expression, indicating that binary classification is misleading since toxicity also depends on activation state._

This observation has small value, given the above references. Additionally, Suau et al. 2024 (already cited) also showed that toxicity is not binary, but depends on magnitudes and signs.

**W2:** L197 _As shown in Figures 2(c), even when completely removing the top 500 most toxic vectors (setting scaling factors to 0), toxicity scores decrease by only 0.08 on GPT2 and 0.04 on Qwen3._

Previous work ([Rodriguez et al. ICLR 2025](https://arxiv.org/abs/2410.23054v1), [Rodriguez et al. NeurIPS 2025](https://arxiv.org/abs/2503.10679)) has shown that relying on scaled vectors suffers from driving activations OOD (with respect to the distributions of activations seen in training). This makes conditioning (eg. toxicity mitigation) lower, and harms utility (eg. MMLU). These works propose to preserve distributions by _transporting_ activations always in-distributions, effectively surpassing ITI, CAA or AurA at toxicity mitigation and utility preservation. These observations make Experiment 2 in the manuscript of less value for the community.

**W3:** _Despite Self-Examination achieving competitive toxicity scores, it lacks understanding of toxic regions within the model and fails to provide interpretable insights into the underlying mechanisms of toxicity generation._

I respectfully disagree. The results in Table 3 for Self-Examination show  similar toxicity results (sometimes better) than GLOSE, while maintaining much better perplexity, fluency and consistency. **For example, for Llama3.1-8B GLOSE incurs a 14% increase of PPL, while Self-Examination keeps the same PPL as the original model**!
In my opinion, Self-Examination behavior is preferable from a user. The comparison is even harder since the table does not include statistical evidence. I encourage the authors to include statistics over 3+ runs.

The paragraph in L409 "GLOSE preserves model capabilities better." should be revisited, since it is misleading the reader. Self-Examination does preserve model capabilities better, the results in Table 3 are clear. This paragraph should be amended with a faithful analysis.

**W4:** Lack of comparison with steering community.

This work deserves a comparison with steering methods, which have shown good results at toxicity mitigation on RealToxicityPrompts. I encourage the authors to include comparison to (some of) these methods: **LinearAcT** [Rodriguez et al. ICLR 2025](https://arxiv.org/abs/2410.23054v1), **LinEAS** [Rodriguez et al. NeurIPS 2025](https://arxiv.org/abs/2503.10679) or **ITI** [Kenneth et al. NeurIPS 2024](https://arxiv.org/abs/2306.03341).  Moreover, **LinEAS** proposes an end-to-end approach that learns an intervention _for all layers at once_ via gradient descent, which makes comparison more suitable to the global approach proposed in this manuscript. Also, LinEAS shows results using only 32 toxic sentences for training, which is far less than the 500 used by GLOSE (unless I misread that number).

**W5.** Metrics such as PPL might not capture the utility of the model to its full extent. I encourage the authors to include additional metrics (eg. MMLU or similar).

**W6.** There are several typos and the overall writing should be revisited. Also, the method is called differently (GLOSS) in some parts of the paper.

**Overall comments:**

Given the above weaknesses, I cannot recommend this paper for acceptance. The results in Table 3 have not been duly analyzed, since Self-examination is showing the best trade-off between toxicity mitigation and utility. Moreover, modern steering methods are lacking as comparison, these methods being important given their good performance at toxicity mitigation and how sample efficient they are.
Additionally, some experiments in the manuscript are oversee previous findings in the community (see W1, W2).

**Questions:**

L127-128: _We select the top-k tokens from the projection of u, offering an interpretable approximation of its
semantic content. Notably, this projection depends only on the direction of u, not its magnitude._

How are the top tokens selected? How are they ranked? And how is $k$ selected?


L137: _we use Detoxify1 to score the toxicity of the first 10 generated tokens for each prompt._

Using only 10 tokens might lead to very biased/noisy results. Toxicity might not be appreciated in the 10 first generated tokens. Longer continuations might become much more toxic. I do not agree with using 10 tokens as evaluation setup, but rather suggest to use 50+.

L148: _Experiment 1: Impact of value vector activations on toxicity expression. Following Lee et al. (2024), we train a linear probe Wtoxic on the Jigsaw dataset to classify toxicity, achieving over 94% accuracy on both models._

Is one probe trained on each layer? What is the shape of $W_{toxic}$?

_We identify toxic and non-toxic vectors by selecting those with the highest and lowest cosine similarity to $W_{toxic}$. _

Selecting _those_, which vectors are used to measure similarity to $W_{toxic}$? These steps require much more explanation for a reader to follow.

_The bad-word list serves only for ranking and can be replaced by any toxicity signal (e.g., classifier scores or implicit bias indicators)._

Have the authors tried these other approaches? What are the benefits of using a BoW with respect to a classifier or bias indicator? Is the BoW flagging words as always toxic while they might be toxic _only_ in specific contexts?



Typos:
* L26: show **~GloSS~ that GLOSE** achieves
* L45: methods (Rafailov et al., 2023) mitigate**~s~**
* L121: E**~quations e~**quation 1 make**s** explicit
* L133: In this section, we conduct **a** systematic analysis
* L134: To pro**~b~v**e it
* L308: 4 DETOXIFICATION METHOD: **~GLOSS~GLOSE**
* Figure 4 says 4 stage procedure. Correct to 3 stages.

---

### Official Review · Reviewer_pSRf · 2025-10-28

**Soundness:** 2
**Presentation:** 1
**Contribution:** 2
**Rating:** 2
**Confidence:** 3

**Summary:**

This paper introduces a method designed to understand and mitigate the generation of toxic content in LLMs. The paper discovered that LLMs contain a shared global toxic subspace across all layers and proposed a method that mitigates toxicity by projecting the FFN weights onto the orthogonal complement of the identified global subspace. Experiments demonstrate that GLOSE achieves high detoxification performance while preserving general capabilities.

**Strengths:**

Through systematic analysis, the paper identifies limitations in previous approaches, specifically noting that toxic vectors in the FFN weights can be reconstructed via linear combinations of non-toxic vectors, necessitating the removal of the entire subspace. Also, layer-wise toxic subspaces, such as ProFS, are unreliable for some layers.

The method consistently achieves superior toxicity reduction results across diverse LLMs (e.g., Qwen, LLama, Gemma) compared to major detoxification baselines, including complex fine-tuning methods like SSFT and DPO, and previous mechanistic methods like ProFS.

**Weaknesses:**

The proposed method, GLOSE, significantly relies on the framework established by ProFS (Uppaal et al., 2025). Specifically, Step 1 (Layer-wise candidate extraction) and the final removal procedure (orthogonal projection onto the FFN weights) appear functionally identical to the core mechanisms introduced in ProFS. The introduction part mentioning the proposed method and the methodological overview do not sufficiently emphasize that GLOSE is fundamentally an extension of ProFS. The paper must explicitly position GLOSE as an enhancement of the layer-wise ProFS approach, using toxicity ranking (Step 2) and PCA (Step 3) as the key novel mechanisms to overcome the empirical limitations of single layer-wise method demonstrated in Section 3.

While the paper offers empirical justification (Section 3) and ablation studies proving that toxicity ranking (Step 2) and PCA (Step 3) are necessary to overcome the limitations of the layer-wise method (ProFS), the approach lacks a robust theoretical justification or mechanistic derivation for selecting these specific statistical tools (ranking and PCA) to synthesize the global subspace. Given that the core mechanisms of subspace identification (contrastive representation via activation differences) and final elimination (orthogonal projection) are shared with ProFS, the incremental contribution provided by the intermediate steps (ranking and PCA) is not strongly substantiated from a theoretical standpoint, potentially leading to an underestimation of the technical contribution relative to the foundational framework established by ProFS.

The toxicity ranking step (Phase 2) is crucial for filtering noise but relies on overlap measurement with a predefined bad words list to quantify toxicity strength. Although the methodology notes that this list could be replaced by other toxicity signals, the reliance on a vocabulary list might potentially limit the method's ability to capture subtle or contextually toxic directions that do not contain explicit profanity.

The paper lacks theoretical grounding for the core components of GLOSE, like the contrastive representation and SVD in Step 1. Since related prior work (ProFS) provided theoretical justification for these choices (e.g., potentially relating them to preference optimization principles like DPO), the absence of theoretical rationale makes the design choices of GLOSE less rigorous and requires the reader to read external references (ProFS) to understand the necessity.

**Questions:**

In ProFS, toxic directions are identified by decomposing the contrastive representation using SVD. The proposed method, GLOSE, further filters these toxic directions by employing a bad words list (Gehman et al., 2020). If the bad words list is utilized, is the step of decomposing the contrastive representation via SVD still necessary? For instance, would it be sufficient to prepare a large number of random directions and filter them solely using the bad words list?

---

### Official Review · Reviewer_CCY2 · 2025-10-29

**Soundness:** 2
**Presentation:** 3
**Contribution:** 1
**Rating:** 2
**Confidence:** 4

**Summary:**

This paper hypothesizes that toxicity in large language models (LLMs) is captured within a specific subspace of their representation space, and that a global toxicity subspace exists in the model’s FFN outputs across layers. To construct this global subspace, the authors first identify layer-wise directions that explain the major variance between toxic and non-toxic representations. These directions are then ranked according to their association with toxic vocabulary. The global toxicity subspace is defined as the set of directions whose association with toxic words exceeds a chosen threshold.

Experimental results suggest that projecting the parameters of the feed-forward networks (FFNs) onto the nullspace of this global toxicity subspace effectively reduces model toxicity.  Although the finding of a global toxicity subspace is interesting, I don't find the detoxification approach proposed in this paper is novel and neither is fairly evaluated against existing methods.

**Strengths:**

1. The paper provides evidence that toxicity is represented within a cross-layer shared subspace in LLMs.
2. It shows that activation-based approaches such as GLOSE and ProFS outperform fine-tuning methods in reducing model toxicity while preserving general performance.

**Weaknesses:**

1. Iterative Nullspace Projection (INLP) [1] is missing in the related work, which also argues that features (e.g., gender bias) occupy a subspace within hidden representations, and a cleaner removal of these features can be achieved by projecting onto the corresponding nullspace. The major difference between GLOSE and INLP seems to be the projection target: GLOSE projects MLP weights, whereas INLP projects hidden representations.
2. Building on this, the paper should compare its method against INLP to clarify the advantages. I am also curious about the performance when directions are derived by applying SVD to the residual stream representations of toxic and non-toxic outputs across layers, followed by the same ranking and selecting processes used for GLOSE.
3. GLOSE ranks the directions used to construct the global subspace according to their association with bad words when projected into the vocabulary space. While this heuristic may effectively capture directions encoding explicit toxicity (e.g., cuss words), it might overlook more covert or context-dependent forms of toxicity, such as hate speech, that are conveyed through longer generations rather than a single token / word.
4. The claim that GLOSE preserves general capabilities is insufficiently supported. The evaluation relies on fluency, consistency, and perplexity metrics on WikiText alone. To substantiate the claim,  the paper should test its approach, at least, on a less contaminated general task benchmark (like MMLU and BBH).
5. In comparing ProFS and GLOSE, the number of projected directions (i.e., subspace dimensionality) does not appear to be controlled. According to Appendix C.3, GLOSE is applied to more layers than ProFS. It's also hard to tell from table 5 whether GLOSE also operates on a larger subspace than ProFS.
6. It is unclear why the authors study the global toxicity subspace only across MLP layers. Since this subspace is derived from the FFN output activations, which are directly added back into the residual stream, the same toxic subspace should also exist there. Therefore, shouldn’t the authors also project the self-attention layer parameters onto the nullspace of the identified toxicity subspace? This design choice is insufficiently motivated.

References:
[1] Ravfogel, Shauli, et al. "Null it out: Guarding protected attributes by iterative nullspace projection." arXiv preprint arXiv:2004.07667 (2020).

**Questions:**

1. In table 5 of Appendix C3, why does GLOSE consistently intervened on more layers than ProFS? Have you tried projecting the toxicity subspaces in MLP on the same layers used in GLOSE?
2. (minor) GLOSE or GLOSS? Both names appear in the main text. Please use a consistent naming for your method.

---

### Note · Authors · 2026-01-04

I have read and agree with the venue's withdrawal policy on behalf of myself and my co-authors.